# Complex Intervention Programs Integrating Multiple Intervention Strategies Were Not More Effective than Active Control Groups: Evidence from Randomized Controlled Trials

**DOI:** 10.3390/bs15111554

**Published:** 2025-11-14

**Authors:** Shoushi Wang, Chunyang Zhang, Jingyuan Huang, Tianyuan Liu, Wei Xu

**Affiliations:** 1Beijing Key Laboratory of Applied Experimental Psychology, National Demonstration Center for Experimental Psychology Education, Faculty of Psychology, Beijing Normal University, No.19 Xinjiekouwai Road, Haidian District, Beijing 100875, China; wangshoushi2001@126.com (S.W.); 202421061043@mail.bnu.edu.cn (J.H.); 202121061110@mail.bnu.edu.cn (T.L.); 2China Academy of Civil Aviation Science and Technology, Beijing 100028, China; zhangchy@mail.castc.org.cn

**Keywords:** online complex psychological intervention, depression, sleep quality, randomized controlled trial (RCT)

## Abstract

**Background:** Evidence-based complex psychological interventions have been widely applied and appear promising in improving emotional and behavioral disturbances such as depression and sleep problems. However, the effectiveness of these complex psychological interventions, particularly in comparison to active control groups, has yet to be confirmed. **Objectives:** To explore the effects of a complex set of intervention strategies on depression and sleep quality, two randomized controlled trials with active control conditions were conducted. **Method:** A total of 97 college students with depressive symptoms were enrolled in study 1 (Intervention = 48; Active control = 49) and participated in the intervention for depression. A total of 110 college students with sleep problems (intervention = 54; active control = 56) participated in the intervention for sleep quality in study 2. College students in the intervention group received a combination intervention program of cognitive behavioral therapy, mindfulness exercises, and positive psychology for depression or sleep quality, while participants in the active control group read popular science articles about mental health. The intervention lasted two weeks. Depression, sleep quality, and related factors were measured before, during, and after the interventions. **Results:** Although both the intervention and active control conditions effectively reduced depression and sleep problems, the trend of change over time was consistent between both groups. Overall, the effects of the intervention group were not better than those of the active control group. **Conclusions:** The effects of short-term, multi-strategy interventions on depression and sleep quality in our study were not better than simply reading articles about mental health. These findings suggest that simply combining evidence-based components does not necessarily produce superior outcomes.

## 1. Introduction

### 1.1. Global Prevalence of Depression and Insomnia

Currently, interventions for depression and sleep problems are the focus of mental health studies worldwide ([3]; [37]). According to the latest systematic review of the global prevalence of depression and insomnia, the prevalence of depression and insomnia was as high as 28.18% and 23.50%, respectively ([50]). In order to alleviate these symptoms, various psychological interventions have been developed and their effectiveness has been validated in both clinical and non-clinical populations.

### 1.2. The Evidence-Based Psychological Intervention for Depression and Sleep Problems

Broadly speaking, psychological intervention strategies for depression and sleep problems share many similarities ([21]; [89]). One of the most widely discussed interventions for depression and sleep quality is cognitive behavioral therapy (CBT)-based approaches, such as the MoodGYM program ([9]) for depression and cognitive behavioral therapy for insomnia (CBT-I; [59]; [62]). Additionally, a meta-analysis revealed that mindfulness-based interventions are equally effective as CBT-based interventions ([23]), both of which are generally effective for improving depression and sleep quality ([28]; [21]; [57]). Besides CBT-based and mindfulness-based interventions, positive psychology interventions have also demonstrated profitable effects on depression and sleep quality ([7]; [47]; [67]).

### 1.3. The Complex Intervention Program Integrating Multiple Intervention Strategies

However, intervention programs relying on a single treatment strategy often face challenges, such as high attrition rates and limited effectiveness ([34]; [63]; [64]). One approach to address these issues is to integrate different intervention methods. Therefore, researchers are beginning to explore the effects of comprehensive intervention programs that incorporate CBT, mindfulness techniques, and positive psychology intervention techniques ([8]; [54]; [77]; [80]). For example, a comprehensive intervention program that incorporates both CBT and mindfulness techniques has been developed ([54]), as well as the ‘Say yes to life’ (SYTL) program, which includes components of positive psychology, CBT, and mindfulness ([8]).

In addition, with the advancement of computer networking and information technology, internet-based and mobile-based treatment programs aligned with the characteristics of psychological interventions have been designed. These programs make treatment more convenient and accessible, allowing a broader range of individuals to receive effective support ([3]; [62]). Therefore, these programs, which integrate various psychological intervention techniques, have been developed into online platforms and are increasingly used to address depression and sleep problems in the general population.

Therefore, integrating multiple strategies appears to offer a solution to the inefficiencies of single interventions. For complex psychological symptoms, combined intervention strategies may be effective for a broader range of issues, as supported by some studies ([18]; [54]; [80]). For instance, compared to the waiting control group, participants who received a web-based intervention program integrating mindfulness exercises and CBT experienced significant reductions in depression and anxiety, as well as notable increases in mindfulness ([54]).

### 1.4. Challenges and Concerns Regarding the Effectiveness of Complex Intervention Programs

Compared to the widespread application of integrated intervention strategies, careful consideration and evidence regarding their effectiveness have been somewhat neglected. When multiple intervention strategies are combined, the issues associated with each strategy may also accumulate. For example, the problem of high attrition rate remains ([80]). The complexity of combined interventions may pose learning challenges for participants experiencing pronounced psychological symptoms, which may contribute to the elevated attrition rates.

Moreover, different intervention strategies may have varying perspectives on psychological issues. Simply integrating them may result in theoretical incompatibilities or even severe conflicts. Therefore, it is questionable whether such combinations can yield optimized outcomes. Some studies demonstrated that integrative intervention strategies showed only modest effects ([25]; [61]). Additionally, these online interventions, which typically combine multiple strategies, are often applied over short durations, potentially limiting their effectiveness ([62]).

Another major limitation of previous relevant studies is that they mostly utilized waitlist control groups rather than active control groups ([33]; [54]; [62]; [80]). This limitation raises concerns about the placebo effect, which cannot be ignored ([86]). Common factors such as fostering hope, setting realistic expectations for change, providing praise, and delivering psychoeducation have been shown to contribute to positive outcomes ([58]). These findings suggest that simple attention and support from mental health workers may drive much of the improvement, while professional interventions provide an additional but smaller benefit. Thus, while integrative interventions, which combine various strategies, are highly popular in the market, it is essential to return to scientific and rational thinking to carefully reassess their true effectiveness.

### 1.5. Current Study

Considering the achievements and potential limitations of existing studies on interventions for depression and sleep problems, the current study aimed to evaluate the actual effectiveness of a program integrating multiple intervention strategies. To achieve this, two separate randomized controlled trials were conducted: the first trial involved an intervention group for depression and an active control group reading mental health articles related to depression, while the second trial involved an intervention group for sleep quality and an active control group reading mental health articles related to sleep quality. This design was employed to address the potential impact of the placebo effect.

The intervention program for the treatment groups incorporated strategies based on CBT, mindfulness, and positive psychology interventions. Interventions for depression integrate CBT, mindfulness, and positive psychology. These three treatments target different symptoms of depression. CBT primarily focuses on patients’ negative, pessimistic, and self-deprecating thoughts, using cognitive restructuring techniques to reconfigure the linkages between thoughts and emotions ([32]; [71]). Mindfulness assists individuals in recognizing the emergence of emotions and thoughts while adopting a non-judgmental and accepting attitude, simultaneously improving patients’ attention issues ([35]; [71]; [68]). In contrast to addressing negative issues, positive psychology centers on enhancing positive feelings and seeking meaning to counteract anhedonia ([4]; [7]; [39]; [70]). Sleep intervention programs employ a combination of CBT-I and mindfulness. Similarly to depression, CBT-I has been established as the first-line treatment for sleep disorders, addressing the cognitive and behavioral issues underlying sleep problems ([11]; [64]). Mindfulness practice is considered to enhance standard CBT-I protocols by addressing emotional and emotion-regulation skills, while also aiding relaxation ([15]).

Given the potential limitations of this complex intervention, this study refrains from making specific hypotheses and instead seeks to explore the effects of these comprehensive interventions on depression, sleep problems, and related factors, in comparison to active control groups.

## 2. Study 1

Study 1 employed a randomized parallel group design to examine whether an online self-help intervention course could effectively alleviate depression among college students. The intervention group participated in a 14-day continuous online depression intervention, while the control group engaged in an active control by reading daily popular science articles related to depression. In addition to using depressive symptoms as the primary outcome measure, this study also included depression stigma, rumination, cognitive flexibility, perceived stress, and pleasurable experience as secondary outcome measures to comprehensively assess the intervention’s effectiveness ([14]; [29]; [41]; [60]; [82]; [81]). The indicators included in this study were all closely related to the occurrence of depression, allowing for a more comprehensive response to the effects of the intervention.

### 2.1. Methods

#### 2.1.1. Participants

A priori power analysis was conducted using G*Power 3.1 software ([20]) to determine the minimum sample size required for detecting a small-to-medium effect size (Cohen’s f = 0.25). With parameters set to α = 0.05 (two-tailed) and power (1-β) = 0.95, the analysis indicated a requirement of 44 participants per group for a between-subjects ANOVA design. Recruitment was implemented through targeted social media advertisements on platforms frequented by college students. Prospective participants expressing interest in the depression intervention were directed to complete an online eligibility screening questionnaire. Inclusion criteria mandated: (1) current enrollment in a college degree program, (2) absence of concurrent participation in other mental health interventions (e.g., psychotherapy, counseling, or psychotropic medication use), and (3) a baseline score on the Self-Rating Depression Scale (SDS; [90]) corresponding to at least mild depressive symptoms (standardized score < 53, calculated as raw score × 1.25; [76]). Eligible participants were randomly assigned to the intervention or control group in a 1:1 ratio. The randomization sequence was generated by the investigator using SPSS 27, employing a block randomization method with block size 4 to ensure group balance.

#### 2.1.2. Procedure

The study used a single-blind design in which participants were unaware of their group assignments. Before intervention, all subjects in the depression intervention group and the control group underwent pre-test (T1). In the pre-test phase, all subjects were assessed for depressive mood, depressive stigma, rumination, perceived stress, temporal pleasurable experience, cognitive flexibility, social support, and life events. On the seventh day after the intervention, all participants were assessed on all questionnaires same as T1 except social support and life events for the mid-test (T2). At the end of the intervention, all participants were assessed on all questionnaires as T1 except social support for post-test (T3). This study was approved by an Ethics Committee of the Faculty of Psychology (IRB Number: 202303130051). In addition, this study has been pre-registered on the OSF (https://doi.org/10.17605/OSF.IO/EBWKA, accessed on 6 March 2023).

#### 2.1.3. Intervention

The depression intervention groups received a comprehensive intervention program. The program included a 14-day online intervention course (administered via Qualtrics), which is divided into course module and practice module. The course module was based on CBT, which included five parts: introduction of depression, awareness of emotion, challenging irrational thinking, CBT exercises and daily dairies. Each day of the course module included an introduction to CBT skills and interactive exercises. The practice module included three components: (1) a photograph-based intervention, (2) “Three Good Things” intervention, based on positive psychological interventions, and (3) mindfulness meditation. In the first seven days of the program, they recorded three good things that happened each day or took photographs for a specific theme (e.g., “the beauty of nature”, “delicious food”), and in the last seven days, they conducted mindfulness exercises continuously. The specific daily schedule of intervention components is shown in Table 1. The researchers distributed links to the intervention content (implemented using the Qualtrics platform) on a daily basis in accordance with the subjects’ progress of use at 19:00. A reminder to complete the day’s content is sent to the user at 21:00 every day.

The active control group read daily scientific articles about mental health, particularly depression. In order to determine the reading status, the subjects were required to answer 1 or 2 questions related to the content of the articles. These articles originate from psychological counseling organizations, universities, or other psychology-related accounts, and are published on the Wechat Public Platform. The specific topics of the articles, including anxiety and depression, anxiety, emotions, etc., is presented in Appendix A. As with the intervention group, the control group also received three reminders daily at the same times. Upon completion of all testing, the control group received the course package containing the full intervention content.

#### 2.1.4. Measures

Depressive symptoms. Depression symptoms were assessed using the Self-Rating Depression Scale (SDS). The scale was developed by [90] ([90]). The Chinese version of SDS was translated by [79] ([79]). The scale consists of 20 questions and is scored on a 4-point Likert scale. In this study, the Cronbach’s alpha for this scale three times was 0.70, 0.88, 0.89.

Depressive stigma. The Chinese version of the Depression Stigma Scale (DSS; [43]) consists of 18 entries and is scored on a 5-point Likert scale, with higher scores indicating higher levels of stigma and discrimination. In this study, the alpha coefficients of the scale were 0.84, 0.87, and 0.90.

Rumination. The Rumination Response Scale (RRS) was developed by [56] ([56]). [24] ([24]) revised the Chinese version of RRS. The scale consists of 22 items with three dimensions, describing responses to depressed mood that are self-focused, symptom-focused, and focused on the possible causes and consequences of dysphoric mood. This scale is rated on a scale of 1–4, with the higher the score, the more serious the rumination. The RRS was proven to have good reliability in this study (Cronbach’s alpha = 0.91, 0.92, 0.93).

Perceived stress. The Perceived Stress Scale-10 (PSS-10; [13]) was used in this study to measure participants’ perceptions of daily stress. The scale consists of 10 items, and participants were asked to assess their perceived level of stress over the past month on a 5-point scale (from “0 = never” to “4 = often”). Higher scores indicate higher levels of perceived stress. The scale has been shown to have good reliability in this study (Cronbach’s alpha = 0.84, 0.85, 0.90).

Temporal pleasurable experience. We used The Temporal Experience of Pleasure Scale (TEPS; [22]) to measure subjects’ ability to perceive pleasure in daily life. [69] ([69]) revised this scale into the Chinese version. The scale consists of 20 questions, each of which describes a scenario where a person can feel pleasure, and a 1–6 point scale is used to evaluate the subject’s perception of the scenario. The higher the score, the more the subject experiences pleasure on a daily basis. In this study, the Cronbach’s alpha for this scale three times was 0.88, 0.86, 0.88.

Cognitive flexibility. The Cognitive Flexibility Inventory (CFI), which was developed by [17] ([17]), was used in this study to assess cognitive flexibility. The revised Chinese version ([78]) of the questionnaire has demonstrated good reliability and validity in past studies, and is suitable for the measurement of cognitive flexibility of college students. The CFI consists of 20 questions, which are divided into two dimensions, alternatives and control, and is rated on a 5-point scale. The higher the total score, the higher the level of cognitive flexibility an individual exhibits. In the present study, the Cronbach’s alpha for the three measurements of CFI was 0.91, 0.92, 0.93.

Social support. This study used the Social Support Rating Scale ([83]) to measure the social support received by the subjects. The scale consists of four dimensions: subjective support, objective support, objective support usage and total social support, with higher total scores indicating stronger social support around the subjects. The scale has been shown to have good reliability and validity in other studies ([44]).

Life events. The Adolescent Self-Rating Life Events Check-list (ASLEC) developed by [46] ([46]), which consists of 27 negative events, was used in this study to explore whether or not each event occurred and the psychological feelings it caused. In this study, the rating period for the pre-test was the most recent year and the post-test was the last two weeks (that is, during the intervention). The reliability of this scale has been validated by the results of previous studies ([45]).

#### 2.1.5. Data Analysis

The study first tested whether common method bias existed and the difference between the scores of the control and intervention groups in the baseline. Then, we conducted linear mixed-model repeated measures (MMRM) ANOVA with measurement occasion as a within-group factor, intervention as a between-group factor, and social support and life events included as covariates.

### 2.2. Results

#### 2.2.1. Participants Enrollment

Recruitment commenced on 23 February 2023, and concluded the same day following rapid enrollment to meet the predetermined sample size target (*N* = 102 per group), as calculated in the a priori power analysis. A total of 195 individuals initially completed the intervention protocol; 81 participants were excluded prior to randomization due to failure to meet inclusion criteria. The remaining 102 eligible participants were allocated in a 1:1 ratio to either the experimental intervention group (*n* = 51) or the active control group (*n* = 51). The intervention group comprised 24 females (47.1%) and 27 males, with a mean age of 20.88 years (SD = 1.90); the active control group included 33 females (64.7%) and 18 males, and had a mean age of 20.50 years (SD = 1.46). The complete participant flow, including screening, exclusion reasons, and final allocation, is delineated in the flowchart (Figure 1).

#### 2.2.2. Common Method Bias for Study 1

The Harman single-factor method was used to test for common method bias ([87]). The results showed that in pre-test, there were 30 factors with eigenvalues greater than 1, and the variance explained by the first factor was 17.22%; in mid-test, there were 29 factors with eigenvalues greater than 1, and the variance explained by the first factor was 22.88%; in post-test, there were 32 factors with eigenvalues greater than 1, and the variance explained by the first factor was 32.93%. The variances explained by the first factor of three times were all less than the 40% threshold, suggesting that there was no serious common method bias in this study.

#### 2.2.3. Baseline Between-Group Comparisons

The baseline scores of descriptive statistics are shown in Table 2. Between-group *t*-tests were performed on the subjects’ pre-test scores for age, depressed mood, cognitive flexibility, rumination, perceived stress, temporal pleasurable experience, depression stigma, life events, and social support. No significant differences were found between the baseline levels of the intervention and control groups for the remaining variables except for the social support score (*t* = −2.029, *p* = 0.045).

#### 2.2.4. Intervention Effect Test of Study 1

A 2 (group) × 3 (time) mixed-model repeated measures ANOVA was conducted on participants’ depression scores (T3 life events and social support as covariates). Results revealed a significant main effect of time (*F*(2, 190) = 9.43, *p* < 0.001, *η*^2^ = 0.760), a non-significant main effect of group (*F*(1, 95) = 2.29, *p* = 0.134), and a non-significant interaction between time and group (*F*(2, 190) = 0.78, *p* = 0.437). The estimated marginal means and 95% confidence intervals for group × time effects are presented in Appendix A. This indicates that both groups exhibited significant decreases in depression scores over time, with large effect sizes ([12]). However, no significant differences were found in both absolute and changes in depression scores between the two groups.

The MMRM were also conducted for the other scores (T3 life events and social support as covariates). All of the main effects of time were significant, the main effect of group was not significant for all but cognitive flexibility, and all interactions were not significant (Specific results are shown in Table 3). For cognitive flexibility, the scores of control group were significantly higher than the intervention group (*F*(1, 95) = 4.34, *p* = 0.010).

### 2.3. Discussion of Study 1

The results of Study 1 showed that the interaction between group and time was not significant for any of the variables, suggesting that the intervention and active control groups followed the same trend across time points. Although both conditions were effective, these improvements were not attributable to group assignment. This result suggests that the intervention group was no more effective than the active control group. However, after the intervention, depression, depressive stereotypes, rumination, and perceived stress were significantly lower in both the intervention and active control groups, whereas cognitive flexibility and temporal pleasurable experiences were significantly higher, suggesting that both groups may be effective for depression improvement, especially for the change in depression reaching large effects size. In addition, there was a significant difference in social support between the two groups of subjects during the initial phase of the intervention, which, although it was included as a covariate, may also have had an impact on the accuracy of the findings.

## 3. Study 2

The purpose of Study 2 was to verify whether an online self-help sleep intervention course could improve sleep problems in a college student population. The experiment implemented an RCT design in which subjects were randomly assigned to either an intervention group or a positive control group. The control group was an active, parallel control group with the same sample size as the intervention group. The intervention group received an online sleep intervention for 14 days. The control group read sleep-related science articles daily. In order to comprehensively verify the improvement of the intervention on sleep problems, this study included sleep procrastination, rumination, perceived stress, and negative affection as secondary indicators in addition to measuring the subjects’ sleep quality ([10]; [19]; [26]; [30]; [51]; [55]). Measuring these variables at the same time can respond to whether interventions can similarly reduce risk factors for sleep problems.

### 3.1. Methods

#### 3.1.1. Participants

Study 2 had the same design as Study 1, with the same sample size requirement of at least 44 participants per group as calculated by G*power. Also, screening and participation process for subjects of Study 2 had essentially the same as Study 1. The only difference in the screening criteria for Study 2 was a score of greater than 11 on the Pittsburgh sleep quality index (PSQI; [6]). The randomization process for Study 2 was also the same as that for Study 1.

#### 3.1.2. Procedure

The procedure for Study 2 was consistent with Study 1 with three tests administered before the start of the intervention (T1), on the seventh day (T2), and at the end of the intervention (T3). The content of each test in Study 2 was the same, including sleep quality, sleep procrastination, rumination, perceived stress, and negative affection. Study 2 also received approval from the Ethics Committee of the Department of Psychology (IRB Number: 202303130051) and was pre-registered on the OSF (https://doi.org/10.17605/OSF.IO/KYV35, accessed on 26 April 2023).

#### 3.1.3. Intervention

The sleep intervention program was based on CBT-I, incorporating mindfulness meditation, and was delivered via the internet (using Qualtrics). The program consisted of four main modules: sleep restriction, sleep hygiene, sleep cognition, and stimulus control. On the first day of the intervention, each participant in the intervention group received a sleep improvement booklet, which provided an overall preview of the intervention. Each day participants completed four modules in sequence. The sleep restriction module helped the user to establish a good routine by keeping a daily log of sleep time and sleep quality. The sleep hygiene module showed users how to promote sleep through questions, including both behaviors before sleep and the sleep environment. The sleep cognition module aimed to change the user’s misperception of sleep problems through cognitive reconstruction. Finally, the stimulus control module established a good physiological response mechanism by limiting the amount of time the user lay in bed, and provided a variety of mindfulness meditation instruction recordings to help the users relax and fall asleep. Links to the program were sent to participants in the intervention group at 19:00 every day. If not completed, an additional reminder will be sent at 22:00.

The active control group of this study read scientific articles on sleep every day. In addition, the subjects were required to answer 1 or 2 questions related to the content of the articles in order to determine the reading status. These articles were consistent with the sources and research outlined in Study 1, covering topics such as sleep trivia, sleep cycle and biological clock. The specific topics of the article are presented in Appendix A. The timing and frequency of content reminders for participants were identical to those in the intervention group. At the end of the post-test, the control group was given the course package with all intervention content.

#### 3.1.4. Measures

***Sleep quality.*** The Pittsburgh Sleep Quality Index (PSQI; [6]), which was revised into Chinese by [46] ([46]), was used in this study to test participants’ sleep problems. It consists of 18 items organized into 7 dimensions, each scored on a 0–3 scale. Higher scores indicate poorer sleep quality. In this study, the alpha coefficients of this scale were 0.64, 0.75, 0.67.

***Sleep procrastination.*** The Bedtime Procrastination Scale (BPS; [40]), a 9-item scale with a 5-point Likert scale, with higher scores indicating more severe procrastination behaviors. BPS was revised into Chinese by [49] ([49]). The Cronbach’s alpha of this scale was 0.87, 0.90, and 0.92 at three time points.

***Rumination.*** The scale used to test rumination was the same as in Study 1. The RRS was proven to have good reliability in this study (Cronbach’s alpha = 0.91, 0.92, 0.93).

***Perceived stress.*** Study 2 also applied PSS-14 to measure subjects’ perceived stress as in Study 1. PSS-14 was revised by Yang and Huang for the Chinese version ([84]). The scale has been shown to have good reliability in this study (Cronbach’s alpha = 0.89, 0.89, 0.90).

***Negative affection.*** The Depression Anxiety and Stress Scale (DASS-21; [48]) was used in this study to measure subjects’ negative affect, revised into Chinese by [72] ([72]). The scale consists of 21 questions on a 4-point scale (0–3). The higher the score, the more severe the emotional problem. In this study, the alpha coefficients for this scale were 0.94, 0.95, 0.95.

#### 3.1.5. Data Analysis

The data analysis methods for Study 2 were the same as those for Study 1.

### 3.2. Results

#### 3.2.1. Participants Enrollment

Recruitment of study 2 also began on 23 February 2023, and was terminated the same day once the target sample size was reached. Finally, a total of 222 participants completed the enrollment questionnaire, and 113 participants met the criteria to be included in the study. The intervention group was made up of 30 females (53.6%) and 26 males, whose average age was 21.15 years (SD = 1.91). The active control group comprised 30 females (52.6%) and 27 males, with a mean age of 21.36 years (SD = 2.08). The study flowchart is shown in Figure 2.

#### 3.2.2. Common Method Bias

The results of common method bias showed that there was no serious common method bias in this study. Specifically, in pre-test, there were 38 factors with eigenvalues greater than 1, and the variance explained by the first factor was 19.40%; in mid-test, there were 38 factors with eigenvalues greater than 1, and the variance explained by the first factor was 19.46%; in post-test, there were 29 factors with eigenvalues greater than 1, and the variance explained by the first factor was 22.08%. The variances explained by the first factor of three times were all less than the 40% threshold ([87]).

#### 3.2.3. Baseline Between-Group Comparisons

At T1, 25 males (46.3%) and 29 females joined the intervention group and 26 males (46.4%) and 30 females attended the control group subjects. The baseline scores of descriptive statistics of study 2 were shown in Table 4. Comparing the baseline scores, the results showed that there was no significant difference between the baseline levels of the intervention and control groups for all variables.

#### 3.2.4. Intervention Effect Test of Study 2

A 2 (group) × 3 (times) MMRM analysis of subjects’ sleep quality scores was conducted. The results are shown in Table 5, which showed a significant main effect of time (*F*(2, 216) = 77.031, *p* < 0.001, *η*^2^ = 0.416), a non-significant main effect of group (*F*(1, 108) = 2.908, *p* = 0.091), and a non-significant interaction effect of time and group (*F*(2, 216) = 0.499 *p* = 0.608). The estimated marginal means and 95% confidence intervals for interaction effects are presented in Appendix A. This result suggests that the improvement of sleep quality in the intervention group was not better than in the control group in this study. However, both groups significantly improved before and after the intervention and reached large effect sizes ([12]).

For the MMRM result of other outcomes, the main effect of time was significant for all (specific results are shown in Table 4). The main effect of the group was significant for rumination (*F*(1, 108) = 4.338 *p* = 0.040), with the control group having significantly higher ruminative thinking than the intervention group. The interaction effect was not significant for all analyses.

### 3.3. Discussion of Study 2

The results of study 2 showed that after the intervention, sleep quality problems, sleep procrastination, rumination, perceived stress and negative affections significantly reduced in the intervention group and the active control group. Both groups reached large effect sizes for improvements in sleep problems, suggesting that both interventions may have been more effective. However, the group × time interactions were non-significant across all variables, consistent with Study 1. These results mean that the effect of the intervention group was not better than the active control group again.

## 4. General Discussion

Through Study 1 and Study 2, the current study found that the effects of a 2-week online, multi-strategy intervention on depression and sleep quality were not better than simply reading articles about mental health. These findings are consistent with previous studies reporting that active control strategies can be as effective as mindfulness-based CBT programs in decreasing the depression of people with epilepsy ([31]).

One possible reason is that the active control itself effectively addressed depression and sleep problems. In both studies, participants in the active control groups read psychoeducational materials on mental health, which served to reduce stigma and help them develop coping strategies for their depression and sleep issues. In fact, these scientific contents are also incorporated into intervention programs—for instance, the first day of the depression program is an introduction of depression. Our findings suggest that psychological science communication, an often-overlooked component of interventions, may actually play a crucial role in their effectiveness. The process of reading these materials possibly facilitated positive cognitive restructuring, contributing to symptom improvement ([53]). Furthermore, the independent learning of mental health knowledge helped meet the students’ needs for autonomy, which is also beneficial for mental health ([65]).

Second, the short two-week period may have been too brief for participants to fully understand CBT, mindfulness, and positive psychology, limiting the intervention’s effect. Several randomized controlled trials of online interventions examining dose–response effects have found that the number of sessions users complete and the duration of usage correlate with improved outcomes, with these benefits persisting into follow-up period ([16]; [42]; [85]). For instance, an online depression intervention (Deprexis), which integrates multiple therapies including CBT, mindfulness, and positive psychology, typically requires 8 to 12 weeks to complete ([52]). Multiple studies have demonstrated that it yields superior therapeutic outcomes compared to conventional treatment ([1]; [2]; [36]; [73]). [75] ([75]) conducted a large-scale randomized controlled trial involving 1721 participants, finding that digital CBT yielded superior outcomes compared to sleep education, with the intervention lasting nine weeks. Previous studies demonstrating effective self-help sleep interventions typically required four weeks or more of intervention ([27]; [74]). Furthermore, most online intervention feature modules completed weekly or freely over a specified number of weeks ([27]; [66]). Our study required participants to study one module daily may impose excessive cognitive load. Although we did not systematically collect user experience data for qualitative analysis, some participants reported issues such as the high frequency of sessions, concepts that were not explained in accessible enough terms, and theoretical explanations that felt tedious. At the same time, the mindfulness exercises and daily journaling components received more positive feedback from some users. This suggests that learning a large volume of CBT material within a short timeframe may indeed create a significant cognitive load, while mindfulness or journaling practices might be more suitable for such intensive practice schedules. Nevertheless, these interpretations require further confirmation through more rigorous research. However, it is also important to note that extending the intervention duration too much could increase the risk of high attrition rates ([80]).

Third, different intervention strategies operate through distinct mechanisms. For instance, CBT focuses on cognitive challenges and behavioral changes, emphasizing the need for transformation ([38]). In contrast, mindfulness exercises promote acceptance and non-reactivity ([5]). These differing, and potentially conflicting, approaches may have led to counteractive effects in the intervention process. The high prevalence of depressive symptoms and sleep problems suggests an urgent need for effective and accessible psychological interventions. The findings of this study imply that future complex intervention designs should be based on solid theoretical foundations, rather than merely combining existing interventions that have shown some efficacy on their own. On the other hand, incorporating artificial intelligence (AI) into these interventions could offer promising possibilities ([88]). AI could facilitate the development of personalized intervention programs that are tailored to the specific causes of an individual’s symptoms, personality traits, and other relevant factors, rather than relying on a uniform approach for all individuals.

The findings of this study have several important practical implications. Firstly, the effectiveness observed in the active control group suggests that disseminating mental health knowledge can play a meaningful role in reducing public depression and improving sleep quality. For individuals with depressive symptoms or insomnia who are unable to engage in systematic interventions, reading accessible mental health articles may still lead to significant improvements. Secondly, developers of popular psychological intervention programs should reassess the effectiveness of their products. It is crucial that intervention designs prioritize the theoretical coherence among different components to prevent participants from confusion and ensure ease of learning. Finally, the present study raises important questions regarding when and how complexity in multi-component interventions adds value. To more clearly delineate the relationships among components, future research could employ a variety of methodological approaches. For instance, factorial or component analysis designs can be used to disentangle synergistic or antagonistic effects and in-depth interviews can gather user feedback.

Several limitations of this study warrant consideration. First, the intervention duration was relatively short, lasting only two weeks. In addition, this study did not conduct follow-up tracking of participants’ usage and improvements after the intervention, thus the long-term effects were not able to be assessed. As a result, it remains unclear whether a longer-term integrative intervention would yield superior outcomes compared to the active control of reading psychoeducational articles. Future research should explore the effects of extended intervention periods to determine if the benefits of integrated approaches become more pronounced over time. Second, the study lacked a waitlist control group and did not include single-component interventions, such as a purely mindfulness-based intervention. Including these controls would allow for a more comprehensive comparison and a more meticulous understanding of the specific effects of integrative interventions. These limitations suggest areas for further research to more definitively assess the efficacy of complex, multi-strategy interventions.

## 5. Conclusions

Both Study 1 and Study 2 found that the complex interventions of CBT, mindfulness practice, and positive psychology were no better than reading mental health science articles in relieving depressive symptoms, improving sleep quality, and related factors, suggesting that the widely used combination of interventions may not be as effective as expected.

## Figures and Tables

**Figure 1 behavsci-15-01554-f001:**
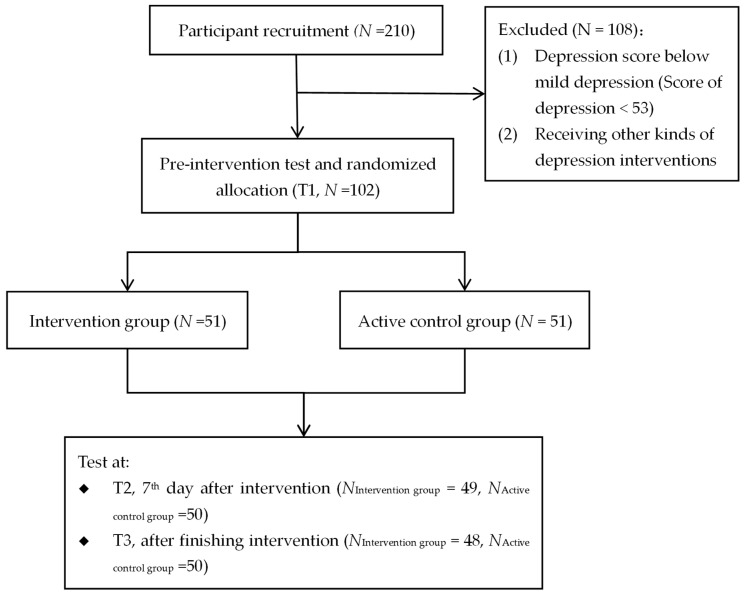
Flowchart of study 1.

**Figure 2 behavsci-15-01554-f002:**
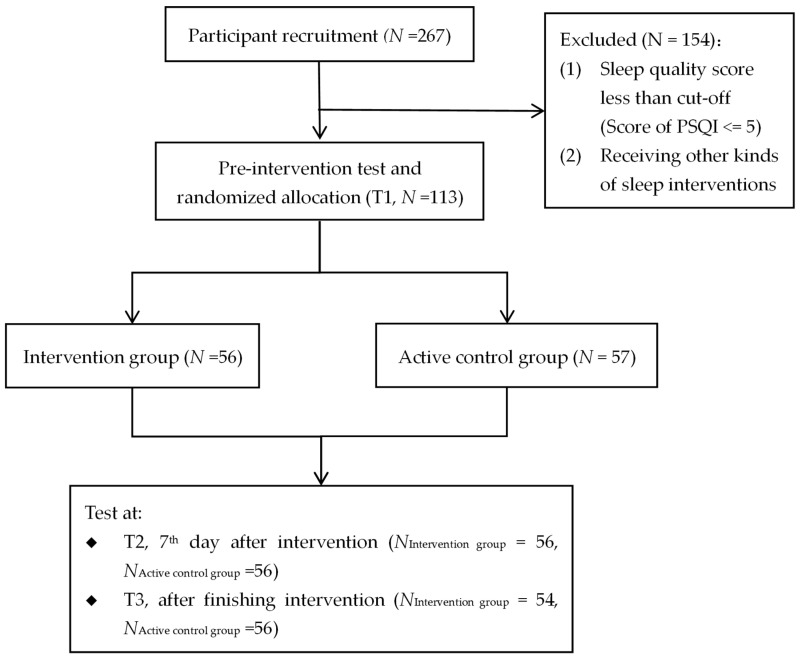
Flowchart of study 2.

**Table 1 behavsci-15-01554-t001:** The specific daily schedule of intervention component of study 1.

Day	Course	Practice
1	Introduction of depression	Photograph-based intervention
2	Introduction of negative emotion	“Three Good Things” intervention
3	Recognizing the emergence of emotions I	Photograph-based intervention
4	Recognizing the emergence of emotions II	“Three Good Things” intervention
5	Assessing the intensity of emotions	Photograph-based intervention
6	Distinguish between thoughts and facts	“Three Good Things” intervention
7	Recognizing Irrational Thinking I	Photograph-based intervention
8	Challenging Irrational Thinking I	Mindfulness meditation
9	CBT exercises I	Mindfulness meditation
10	CBT exercises II	Mindfulness meditation
11	CBT exercises III	Mindfulness meditation
12	CBT exercises IV	Mindfulness meditation
13	Daily dairy	Mindfulness meditation
14	Daily dairy	Mindfulness meditation

**Table 2 behavsci-15-01554-t002:** Baseline score descriptive statistics of study 1.

	Group	*M*	*SD*
Age	Intervention	20.88	1.90
	control	20.50	1.46
Depression	Intervention	62.37	5.89
	control	61.65	6.21
Cognitive flexibility	Intervention	55.61	10.00
	control	58.24	10.85
Rumination	Intervention	63.59	10.56
	control	63.33	10.32
Perceived stress	Intervention	25.75	5.52
	control	25.61	6.11
Temporal pleasurable experience	Intervention	75.96	12.88
	control	81.08	15.50
Depression stigma	Intervention	35.41	10.77
	control	38.12	10.03
Life experience	Intervention	80.22	24.91
	control	83.43	22.72
Social support	Intervention	22.65	5.38
	control	25.04	6.48

**Table 3 behavsci-15-01554-t003:** Results of MMRM analyses of study 1.

		T1 (M ± SD)	T2 (M ± SD)	T3 (M ± SD)	F (Time)	*η*^2^ (Time)	F (Group)	F (Time × Group)
Depression	Intervention	62.42 ± 5.65	46.13 ± 9.07	42.79 ± 9.29	9.25 ***	0.092	2.29	0.78
	Control	61.59 ± 6.00	43.43 ± 8.70	38.73 ± 8.51				
DS	Intervention	35.42 ± 11.01	34.08 ± 12.95	34.17 ± 13.38	11.79 ***	0.113	1.83	2.53
	Control	38.45 ± 9.84	37.16 ± 9.72	34.73 ± 12.34				
Rumination	Intervention	63.65 ± 10.69	56.13 ± 10.52	52.10 ± 11.75	14.86 ***	0.138	1.43	2.13
	Control	63.08 ± 10.14	52.31 ± 11.49	47.37 ± 12.31				
CF	Intervention	55.23 ± 10.13	56.69 ± 8.49	62.06 ± 10.90	8.48 ***	0.084	4.34 *	1.51
	Control	57.88 ± 10.80	62.47 ± 11.54	67.90 ± 12.16				
PS	Intervention	26.00 ± 5.50	21.69 ± 5.43	20.15 ± 6.46	16.25 ***	0.149	0.97	2.21
	Control	25.59 ± 5.89	20.27 ± 5.77	17.34 ± 7.00				
TPE	Intervention	75.65 ± 13.21	79.54 ± 12.78	84.63 ± 14.50	12.43 ***	0.118	2.6	0.03
	Control	81.29 ± 15.65	83.35 ± 13.36	89.76 ± 10.80				

*Note.* DS: depression stigma. CF: cognitive flexibility. PS: perceive stress. TPE: temporal pleasurable experience. * *p* < 0.05, *** *p* < 0.001.

**Table 4 behavsci-15-01554-t004:** Baseline score descriptive statistics of study 2.

	Group	*M*	*SD*
Age	Intervention	21.15	1.91
	Control	21.36	2.08
Sleep quality	Intervention	10.92	2.79
	Control	11.51	2.97
Sleep procrastination	Intervention	35.37	6.58
	Control	34.91	5.68
Rumination	Intervention	51.87	13.40
	Control	55.48	11.09
Perceive stress	Intervention	41.96	11.02
	Control	43.73	7.94
Negative affect	Intervention	22.01	14.53
	Control	25.33	11.26

**Table 5 behavsci-15-01554-t005:** Results of MMRM analyses of study 2.

		T1 (M ± SD)	T2 (M ± SD)	T3 (M ± SD)	F (Time)	*η*^2^ (Time)	F (Group)	F (Time × Group)
SQP	Intervention	10.93 ± 2.79	9.04 ± 2.62	7.54 ± 2.46	77.03 ***	0.416	2.91	0.50
	control	11.52 ± 2.97	9.59 ± 2.79	8.55 ± 2.54				
SP	Intervention	35.37 ± 0.82	29.98 ± 0.94	28.55 ± 1.01	5.43 **	0.274	0.93	1.84
	control	34.91 ± 0.81	31.81 ± 0.92	30.22 ± 0.99				
Rumination	Intervention	51.87 ± 1.68	47.83 ± 1.63	45.39 ± 1.53	4.35 *	0.170	4.30 *	0.44
	control	55.48 ± 1.65	51.64 ± 1.60	50.52 ± 1.50				
PS	Intervention	41.96 ± 1.30	38.44 ± 1.29	35.81 ± 1.26	3.14 *	0.204	2.00	0.40
	control	43.73 ± 1.28	40.36 ± 1.27	38.82 ± 1.24				
NA	Intervention	22.02 ± 1.77	16.94 ± 1.72	13.98 ± 1.61	52.72 ***	0.233	1.87	0.03
	control	25.34 ± 1.74	18.43 ± 1.68	17.64 ± 1.58				

*Note.* SQ: sleep quality problem. SP: sleep procrastination. PS: perceive stress. NA: negative affect. * *p* < 0.05, ** *p* < 0.01, *** *p* < 0.001.

## Data Availability

The data for this study has been publicly released to Mendeley Data (doi: 10.17632/jh3xmgr44y.2).

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
