# Peer review of "Complex Intervention Programs Integrating Multiple Intervention Strategies Were Not More Effective than Active Control Groups: Evidence from Randomized Controlled Trials"

_behavsci, 2025, doi:10.3390/bs15111554_

Round 1

Reviewer 1 Report

Comments and Suggestions for Authors

Strengths:

1.This study addresses an innovative research topic.
2.The manuscript is written with clarity and fluency throughout.
3.The criteria for identifying participants with depressive tendencies are clearly defined—for instance, individuals scoring below 53 were excluded from participation.
4.The authors provided a clear explanation for the lack of statistically significant results, attributing it primarily to the short duration of the intervention. This point is appropriately addressed in the Discussion section.
5.The authors also offered concrete suggestions for future research directions, which are commendable and worthy of encouragement.

In the final two paragraphs of the Discussion section, the authors clearly acknowledge that the two-week duration of the intervention may have been a key factor contributing to the absence of significant differences between the study groups. While this explanation is reasonable, the discussion as a whole appears to lack sufficient citation of relevant literature to support this point.

It is therefore recommended that the authors consult and incorporate additional peer-reviewed literature addressing the effects of longer-duration interventions on similar outcomes. Prior studies have often employed extended intervention periods, and a review of such literature could provide a valuable comparative framework for interpreting the present findings. Discussing the typical length and structure of interventions reported in previous research—and how these differ from the two-week protocol adopted in this study—would enhance the depth and clarity of the discussion. Integrating these references would also help contextualize the non-significant results and offer a stronger rationale for future research directions.

Given that a considerable body of literature exists on this topic, strengthening this section with more comprehensive references would enhance the credibility and depth of the discussion. This recommendation is offered as a constructive suggestion to improve the manuscript.

Comments on the Quality of English Language

ok

Author Response

Reviewer 1

Thank you very much for taking the time to review this manuscript. Please find the detailed responses below and the corresponding revisions in the re-submitted files

Comments 1:

Strengths:

1.This study addresses an innovative research topic.

2.The manuscript is written with clarity and fluency throughout.

3.The criteria for identifying participants with depressive tendencies are clearly defined—for instance, individuals scoring below 53 were excluded from participation.

4.The authors provided a clear explanation for the lack of statistically significant results, attributing it primarily to the short duration of the intervention. This point is appropriately addressed in the Discussion section.

5.The authors also offered concrete suggestions for future research directions, which are commendable and worthy of encouragement.

Response 1: Thank you very much for your recognition of this article.

Comments 2: In the final two paragraphs of the Discussion section, the authors clearly acknowledge that the two-week duration of the intervention may have been a key factor contributing to the absence of significant differences between the study groups. While this explanation is reasonable, the discussion as a whole appears to lack sufficient citation of relevant literature to support this point.

It is therefore recommended that the authors consult and incorporate additional peer-reviewed literature addressing the effects of longer-duration interventions on similar outcomes. Prior studies have often employed extended intervention periods, and a review of such literature could provide a valuable comparative framework for interpreting the present findings. Discussing the typical length and structure of interventions reported in previous research—and how these differ from the two-week protocol adopted in this study—would enhance the depth and clarity of the discussion. Integrating these references would also help contextualize the non-significant results and offer a stronger rationale for future research directions.

Given that a considerable body of literature exists on this topic, strengthening this section with more comprehensive references would enhance the credibility and depth of the discussion. This recommendation is offered as a constructive suggestion to improve the manuscript

Response 2: Agree. We expanded the discussion section to explain the potential impact of intervention duration on effectiveness. Additionally, the duration and scheduling of two previously effective interventions were added into the article (Page 15–16, Line 483–498, red words).

Revision: Several randomised controlled trials of online interventions examining dose-response effects have found that the number of sessions users complete and the duration of usage correlate with improved outcomes, with these benefits persisting into follow-up period (de Graaf et al., 2009; Li et al., 2022; Yang et al., 2024). For instance, an online depression intervention (Deprexis), which integrates multiple therapies including cognitive behavioral therapy, mindfulness, and positive psychology, typically requires 8 to 12 weeks to complete (Meyer & Garcia‐Roberts, 2007). Multiple studies have demonstrated that it yields superior therapeutic outcomes compared to conventional treatment (Beevers et al., 2017; Berger et al., 2018; Klein et al., 2016; Twomey et al., 2017). Vedaa et al. (2020) conducted a large-scale randomised controlled trial involving 1,721 participants, finding that digital cognitive behavioral therapy yielded superior outcomes compared to sleep education, with the intervention lasting nine weeks. Previous studies demonstrating effective self-help sleep interventions typically required four weeks or more of intervention (Ho et al., 2015; van & Cuijpers, 2009). Furthermore, most online intervention feature modules completed weekly or freely over a specified number of weeks (Ho et al., 2015; Saddichha et al., 2014). Our study required participants to study one module daily may impose excessive cognitive load.

Reviewer 2 Report

Comments and Suggestions for Authors

This study investigates the effects of a complex psychological intervention integrating CBT, mindfulness, and positive psychology on depression and sleep quality among college students, compared to an active control group reading psychoeducational articles. The research is well-designed, methodologically sound, and addresses an important question regarding the efficacy of multi-component interventions. The finding that the complex intervention was not superior to the active control is both surprising and significant.

There are some concerns and suggestions:

  1. Short intervention duration: A 2-week period may be insufficient to observe the full benefits of complex interventions. This should be emphasized in the discussion.

  2. Lack of mechanistic explanation: The study does not explore why the combined intervention was not superior. Adding process variables or qualitative feedback could be informative.

  3. Control group content not detailed: The content, source, and quality of the psychoeducational articles are not described, which may affect interpretability.

  4. Language polishing needed: Some sentences are awkwardly phrased. Professional editing by a native English speaker is recommended.

Author Response

Thank you very much for taking the time to review this manuscript. Please find the detailed responses below and the corresponding revisions in the re-submitted files

Comment 1: Short intervention duration: A 2-week period may be insufficient to observe the full benefits of complex interventions. This should be emphasized in the discussion.

Response 1: Agree. We expanded the discussion section to explain the potential impact of intervention duration on effectiveness. Additionally, the duration and scheduling of two previously effective interventions were added into the article (Page 15–16, Line 483–498, red words).

Revision: Several randomised controlled trials of online interventions examining dose-response effects have found that the number of sessions users complete and the duration of usage correlate with improved outcomes, with these benefits persisting into follow-up period (de Graaf et al., 2009; Li et al., 2022; Yang et al., 2024). For instance, an online depression intervention (Deprexis), which integrates multiple therapies including cognitive behavioral therapy, mindfulness, and positive psychology, typically requires 8 to 12 weeks to complete (Meyer & Garcia‐Roberts, 2007). Multiple studies have demonstrated that it yields superior therapeutic outcomes compared to conventional treatment (Beevers et al., 2017; Berger et al., 2018; Klein et al., 2016; Twomey et al., 2017). Vedaa et al. (2020) conducted a large-scale randomised controlled trial involving 1,721 participants, finding that digital cognitive behavioral therapy yielded superior outcomes compared to sleep education, with the intervention lasting nine weeks. Previous studies demonstrating effective self-help sleep interventions typically required four weeks or more of intervention (Ho et al., 2015; van & Cuijpers, 2009). Furthermore, most online intervention feature modules completed weekly or freely over a specified number of weeks (Ho et al., 2015; Saddichha et al., 2014). Our study required participants to study one module daily may impose excessive cognitive load.

Comment 2: Lack of mechanistic explanation: The study does not explore why the combined intervention was not superior. Adding process variables or qualitative feedback could be informative.

Response 2: Since all our research variables have already been presented in the study and we cannot add new variables, it is not possible to supplement the analysis of process variables. We did not systematically collect qualitative responses. However, we did receive some user feedback, which we have incorporated into the discussion section (Page 16, Line 498–506, red words).

Revision: Although we did not systematically collect user experience data for qualitative analysis, some participants reported issues such as the high frequency of sessions, concepts that were not explained in accessible enough terms, and theoretical explanations that felt tedious. At the same time, the mindfulness exercises and daily journaling components received more positive feedback from some users. This suggests that learning a large volume of cognitive behavioral therapy material within a short timeframe may indeed create a significant cognitive load, while mindfulness or journaling practices might be more suitable for such intensive practice schedules. However, these inferences still require verification through more rigorous research.

Comment 3: Control group content not detailed: The content, source, and quality of the psychoeducational articles are not described, which may affect interpretability.

Response 3: Thank you for your reminder. We have incorporated details regarding the sources of the control group articles into the Methods section, and added Table 2 to specify the particular topics of the articles (Page 6, Lines 207-211; Page 11, Lines 382–385, red words).

Revision 3.1: These articles originate from psychological counselling organizations, universities, or other psychology-related accounts, and are published on the Wechat Public Platform. The specific topics of the articles, including anxiety and depression, anxiety, emotions, etc., is presented in Table S1 (supplementary material).

Revision 3.2: These articles were consistent with the sources and research outlined in Study 1, covering topics such as sleep trivia, sleep cycle and biological clock. The specific topics of the article are presented in Table S1 (supplementary material).

Table S1 Scientific articles topics for active control groups

Day

Study 1

Study 2

1

emotionally sensitive

sleep trivia

2

compassion fatigue

sleep cycle

3

anxiety

sleep issues

4

accepting depression

biological clock

5

emotional suppression schema

duration vs quality

6

depression

difficulty falling asleep

7

complex post-traumatic stress disorder (CPTSD)

late-night procrastination

8

anxiety

addressing procrastination

9

depression

pre-sleep arousal

10

automated thinking

sleep myths

11

anxiety

how to sleep better

12

the therapeutic effects of melancholic music

bedtime routine

13

how reason and emotion coexist

bedtime routine 2

14

the therapeutic effects of music

advanced techniques

Common 4:

Language polishing needed: Some sentences are awkwardly phrased. Professional editing by a native English speaker is recommended.

Response 4:  Thank you for the reminder. We have polished the language throughout.

Reviewer 3 Report

Comments and Suggestions for Authors

General evaluation

This manuscript reports two randomized controlled trials evaluating short-term (14-day) online complex interventions integrating CBT, mindfulness, and positive psychology strategies, compared with active psychoeducation control groups. Both studies address important clinical topics—depression and sleep problems—and share a structured methodology. The finding that complex multi-component interventions did not outperform active controls is intriguing and valuable for intervention science. However, despite the methodological clarity, several conceptual, design, and analytic issues limit the interpretability of the null findings. Most importantly, the theoretical justification for combining heterogeneous components (CBT, mindfulness, and positive psychology) is insufficient, and the mechanisms through which these components are expected to interact or synergize are not well articulated.

Major comments

  1. The manuscript needs a clearer theoretical rationale for combining CBT, mindfulness, and positive psychology. These approaches rely on distinct mechanisms—cognitive restructuring, nonjudgmental awareness, and positive affect enhancement—and may not be additive in such a brief format. The authors should clarify whether the integration was theory-driven or pragmatic. It is also possible that the intervention’s complexity diluted the effectiveness of each component, resulting in a blurred or unfocused application of the techniques. This issue warrants further discussion.
  2. The control group received psychoeducation content, which can itself produce therapeutic benefit through psychoeducation and expectancy, which are already the effective intervention components.
  3. The MMRM analyses are appropriate, but additional transparency is needed(for both studies). Please (a) provide estimated marginal means and 95% confidence intervals for group × time effects; (b) include effect sizes for key outcomes.
  4. Study 2 lacks basic demographic data, particularly gender distribution.
  5. The study raises valuable questions about when complexity adds value. The Discussion should explicitly recommend methodological approaches—such as factorial or component analysis designs—to disentangle additive versus synergistic effects of CBT, mindfulness, and positive psychology elements. This would make the research’s contribution more forward-looking.

Minor comments

  1. The manuscript would benefit from careful language polishing by a fluent English speaker or professional editing service. Some sentences are overly long, contain redundant background material, or use awkward phrasing. Improving readability will help highlight the study’s strengths.
  2. CONSORT flow and randomization details.Specify block size and confirm that participant numbers in flowcharts match those in the main text.
  3. Since daily reminders were sent, report whether reminder frequency influenced adherence or outcomes.
  4. The lack of long-term follow-up is a limitation; if any delayed post-test data exist, they should be included.

Author Response

General evaluation

This manuscript reports two randomized controlled trials evaluating short-term (14-day) online complex interventions integrating CBT, mindfulness, and positive psychology strategies, compared with active psychoeducation control groups. Both studies address important clinical topics—depression and sleep problems—and share a structured methodology. The finding that complex multi-component interventions did not outperform active controls is intriguing and valuable for intervention science. However, despite the methodological clarity, several conceptual, design, and analytic issues limit the interpretability of the null findings. Most importantly, the theoretical justification for combining heterogeneous components (CBT, mindfulness, and positive psychology) is insufficient, and the mechanisms through which these components are expected to interact or synergize are not well articulated.

Response: Thank you very much for taking the time to review this manuscript. Please find the detailed responses below and the corresponding revisions in the re-submitted files

Major comments

Comment 1: The manuscript needs a clearer theoretical rationale for combining CBT, mindfulness, and positive psychology. These approaches rely on distinct mechanisms—cognitive restructuring, nonjudgmental awareness, and positive affect enhancement—and may not be additive in such a brief format. The authors should clarify whether the integration was theory-driven or pragmatic. It is also possible that the intervention’s complexity diluted the effectiveness of each component, resulting in a blurred or unfocused application of the techniques. This issue warrants further discussion.

Response 1: Our selection of intervention methods was primarily guided by whether they are widely implemented and supported by empirical research, and whether different methods target distinct symptoms. We have supplemented the rationale for method selection in the introduction. Moreover, we have also supplemented our discussion with the potential cognitive load that short-term interventions may entail  (Page 3-4, Lines 124-141; Page 16, Lines 495–498, red words).

Revision 1: The intervention program for the treatment groups incorporated strategies based on CBT, mindfulness, and positive psychology interventions. Interventions for depression integrate CBT, mindfulness, and positive psychology. These three treatments target different symptoms of depression. CBT primarily focuses on patients' negative, pessi-mistic, and self-deprecating thoughts, reconstructing the link between thoughts and emotions (Hundt et al., 2013; Sverre et al., 2023). Mindfulness assists individuals in recognising the emergence of emotions and thoughts while adopting an accepting attitude, simultaneously improving patients' attention issues (Klainin-Yobas et al., 2012; Sverre et al., 2023; Sevilla-Llewellyn-Jones et al., 2018). In contrast to addressing nega-tive issues, positive psychology centres on enhancing positive feelings and seeking meaning to counteract anhedonia (Bolier et al., 2013; Carr et al., 2011; Koydemir et al., 2021; Steger et al., 2014). Sleep intervention programmes employ a combination of CBT-I and mindfulness. Similar to depression, CBT-I has been established as the first-line treatment for sleep disorders, addressing the cognitive and behavioural issues under-lying sleep problems (Cleary et al., 2024; Ritterband et al., 2017). Mindfulness practice is considered to enhance standard CBT-I protocols by addressing emotional and emo-tion-regulation skills, while also aiding relaxation (de Entrambasaguas et al., 2013).

Revision 2: Furthermore, most online intervention feature modules completed weekly or freely over a specified number of weeks (Ho et al., 2015; Saddichha et al., 2014). Our study required participants to study one module daily may impose excessive cognitive load.

Comment 2: The control group received psychoeducation content, which can itself produce therapeutic benefit through psychoeducation and expectancy, which are already the effective intervention components.

Response 2: Agreed. We have further emphasised this point in the discussion section (Page 14, Lines 469–480, red words).

Revision:One possible reason is that the active control itself effectively addressed depression and sleep problems. In both studies, participants in the active control groups read psychoeducational materials on mental health, which served to reduce stigma and help them develop coping strategies for their depression and sleep issues. In fact, these scientific contents are also incorporated into intervention programs—for instance, the first day of the depression program is an introduction of depression. Our findings suggest that psychological science communication, an often-overlooked component of interventions, may actually play a crucial role in their effectiveness. The process of reading these materials possibly facilitated positive cognitive restructuring, contributing to symptom improvement (Mhango et al., 2023). Furthermore, the independent learning of mental health knowledge helped meet the students' needs for autonomy, which is also beneficial for mental health (Ryan & Deci, 2017).

Comment 3:The MMRM analyses are appropriate, but additional transparency is needed(for both studies). Please (a) provide estimated marginal means and 95% confidence intervals for group × time effects; (b) include effect sizes for key outcomes.

Response 3: Thank you for your suggestion. We have included two tables containing all estimated marginal means and 95% confidence intervals for group × time effects in the supplementary materials (Table S3 and Table S3). Given our focus on intervention effects and the absence of significant interactions, we have added effect sizes for all time main effects in Table 3 (Page 9) and Table 5 (Page 13). In the results section, we specifically highlighted the effect sizes for the time main effects of the primary variables (depression, sleep quality problems).

Revisions:

Table 3 Results of MMRM analyses of study 1

T1

(M±SD)

T2

(M±SD)

T3

(M±SD)

F(Time)

η2 (Time)

F(Group)

F(Time × Group)

Depression

Intervention

62.42±5.65

46.13±9.07

42.79±9.29

9.25***

0.092

2.29

0.78

Control

61.59±6.00

43.43±8.70

38.73±8.51

DS

Intervention

35.42±11.01

34.08±12.95

34.17±13.38

11.79***

0.113

1.83

2.53

Control

38.45±9.84

37.16±9.72

34.73±12.34

Rumination

Intervention

63.65±10.69

56.13±10.52

52.10±11.75

14.86***

0.138

1.43

2.13

Control

63.08±10.14

52.31±11.49

47.37±12.31

CF

Intervention

55.23±10.13

56.69±8.49

62.06±10.90

8.48***

0.084

4.34*

1.51

Control

57.88±10.80

62.47±11.54

67.90±12.16

PS

Intervention

26.00±5.50

21.69±5.43

20.15±6.46

16.25***

0.149

0.97

2.21

Control

25.59±5.89

20.27±5.77

17.34±7.00

TPE

Intervention

75.65±13.21

79.54±12.78

84.63±14.50

12.43***

0.118

2.6

0.03

Control

81.29±15.65

83.35±13.36

89.76±10.80

Note. DS: depression stigma. CF: cognitive flexibility. PS: perceive stress. TPE: temporal pleasurable experience. *p<0.05, ***p<0.001.

Table 5 Results of MMRM analyses of study 2

T1

(M±SD)

T2

(M±SD)

T3

(M±SD)

F(Time)

η2 (Time)

F(Group)

F(Time × Group)

SQP

Intervention

10.93±2.79

9.04±2.62

7.54±2.46

77.03***

0.416

2.91

0.50

control

11.52±2.97

9.59±2.79

8.55±2.54

SP

Intervention

35.37±0.82

29.98±0.94

28.55±1.01

5.43**

0.274

0.93

1.84

control

34.91±0.81

31.81±0.92

30.22±0.99

Rumination

Intervention

51.87±1.68

47.83±1.63

45.39±1.53

4.35*

0.170

4.30*

0.44

control

55.48±1.65

51.64±1.60

50.52±1.50

PS

Intervention

41.96±1.30

38.44±1.29

35.81±1.26

3.14*

0.204

2.00

0.40

control

43.73±1.28

40.36±1.27

38.82±1.24

NA

Intervention

22.02±1.77

16.94±1.72

13.98±1.61

52.72***

0.233

1.87

0.03

control

25.34±1.74

18.43±1.68

17.64±1.58

Table S2 Estimated marginal means and 95% confidence intervals for group × time effects for Study 1.

Group

Time

M

SD

95% CI

Lower

Upper

Depression

Intervention

1

62.303

0.847

60.620

63.985

2

45.662

1.249

43.181

48.143

3

42.210

1.198

39.831

44.590

Control

1

61.704

0.838

60.039

63.368

2

43.882

1.236

41.427

46.337

3

39.304

1.186

36.950

41.658

Cognitive flexibility

Intervention

1

55.792

1.484

52.845

58.739

2

57.182

1.451

54.300

60.064

3

62.643

1.605

59.455

65.830

control

1

57.326

1.468

54.410

60.242

2

61.985

1.436

59.133

64.837

3

67.330

1.588

64.176

70.484

Rumination

Intervention

1

55.792

1.484

52.845

58.739

2

57.182

1.451

54.300

60.064

3

62.643

1.605

59.455

65.830

control

1

57.326

1.468

54.410

60.242

2

61.985

1.436

59.133

64.837

3

67.330

1.588

64.176

70.484

Perceive stress

Intervention

1

25.620

0.794

24.043

27.197

2

21.472

0.798

19.888

23.056

3

19.850

0.926

18.011

21.689

control

1

25.964

0.786

24.404

27.525

2

20.476

0.789

18.909

22.044

3

17.637

0.916

15.817

19.457

Temporal pleasurable experience

Intervention

1

76.720

1.927

72.893

80.547

2

79.871

1.911

76.075

83.667

3

85.294

1.810

81.700

88.888

control

1

80.234

1.907

76.447

84.020

2

83.025

1.891

79.268

86.781

3

89.100

1.791

85.543

92.656

Depression stigma

Intervention

1

34.913

1.494

31.946

37.880

2

33.620

1.628

30.387

36.853

3

34.127

1.835

30.484

37.771

control

1

38.942

1.478

36.007

41.878

2

37.617

1.611

34.418

40.817

3

34.773

1.816

31.168

38.378

Table S3 Estimated marginal means and 95% confidence intervals for group × time effects for Study 2.

Group

Time

M

SD

95% CI

Lower

Upper

Sleep quality problem

Intervention

1

10.926

0.393

10.147

11.704

2

9.037

0.368

8.307

9.767

3

7.537

0.340

6.863

8.212

Control

1

11.518

0.386

10.753

12.282

2

9.589

0.362

8.872

10.306

3

8.554

0.334

7.891

9.216

Negative affect

Intervention

1

22.019

1.766

18.518

25.519

2

16.944

1.711

13.553

20.336

3

13.981

1.604

10.803

17.160

control

1

25.339

1.734

21.902

28.777

2

18.429

1.680

15.098

21.759

3

17.643

1.575

14.521

20.764

Rumination

Intervention

1

51.870

1.671

48.558

55.183

2

47.833

1.623

44.617

51.050

3

45.389

1.520

42.375

48.403

control

1

55.482

1.641

52.229

58.735

2

51.643

1.593

48.485

54.801

3

50.518

1.493

47.559

53.477

Perceive stress

Intervention

1

51.870

1.671

48.558

55.183

2

47.833

1.623

44.617

51.050

3

45.389

1.520

42.375

48.403

control

1

55.482

1.641

52.229

58.735

2

51.643

1.593

48.485

54.801

3

50.518

1.493

47.559

53.477

Sleep procrastination

Intervention

1

35.370

0.836

33.713

37.028

2

29.981

0.949

28.101

31.862

3

28.556

1.015

26.544

30.567

control

1

34.911

0.821

33.283

36.538

2

31.804

0.932

29.957

33.650

3

30.214

0.996

28.239

32.189

Comment 4: Study 2 lacks basic demographic data, particularly gender distribution.

Response 4: Thank you for the reminder. We have added descriptions of gender and age to the first part of the results (Page 7, Lines 278–281; Page 12, Lines 416–418, red words).

Revision 4.1: The intervention group comprised 24 females (47.1%) and 27 males, with a mean age of 20.88 years (SD=1.90); the active control group included 33 females (64.7%) and 18 males, and had a mean age of 20.50 years (SD=1.46).

Revision 4.2: The intervention group was made up of 30 females (53.6%) and 26 males, whose average age was 21.15 years (SD = 1.91). The active control group comprised 30 females (52.6%) and 27 males, with a mean age of 21.36 years (SD = 2.08).

Comment 5: The study raises valuable questions about when complexity adds value. The Discussion should explicitly recommend methodological approaches—such as factorial or component analysis designs—to disentangle additive versus synergistic effects of CBT, mindfulness, and positive psychology elements. This would make the research’s contribution more forward-looking.

Response 5: Thank you for your suggestion. We have addressed this in the discussion section and incorporated the other two recommended approaches for future research  (Page 16, Lines 531–536, red words).

Revision: Finally, the present study raises important questions regarding when and how complexity in multi-component interventions adds value. To more clearly delineate the relationships among components, future research could employ a variety of methodological approaches. For instance, factorial or component analysis designs can be used to disentangle syn-ergistic or antagonistic effects; in-depth interviews can gather user feedback; and in-tensive longitudinal data can be collected to capture the immediate impacts of different modules.

Minor comments

Comment 1: The manuscript would benefit from careful language polishing by a fluent English speaker or professional editing service. Some sentences are overly long, contain redundant background material, or use awkward phrasing. Improving readability will help highlight the study’s strengths.

Response 1:  Thank you for the reminder. We have polished the language throughout.

Comment 2: CONSORT flow and randomization details. Specify block size and confirm that participant numbers in flowcharts match those in the main text.

Response 2: We have added a description of the block size in the Methods section and verified the number of participants in all sections (Page 4, Lines 172–174, red words).

Revision: Eligible participants were randomly assigned to the intervention or control group in a 1:1 ratio. The randomization sequence was generated by the investigator using SPSS, employing a block randomization method with block size 4 to ensure group balance.

Comment 3: Since daily reminders were sent, report whether reminder frequency influenced adherence or outcomes.

Response 3: Since both studies had intervention and control groups receiving the same number and timing of completion reminders daily, it was not possible to compare the impact of reminders on adherence or outcomes within this study. We added instructions regarding reminders to the intervention protocol for the control group (Page 5, Lines 211–212; Page 11, Lines 385–386, red words).

Revision 1:As with the intervention group, the control group also received three reminders daily at the same times.

Revision 2:The timing and frequency of content reminders for participants were identical to those in the intervention group.

Comment 4: The lack of long-term follow-up is a limitation; if any delayed post-test data exist, they should be included.

Response 4: This study did not conduct follow-up tracking of the intervention's effects or usage. This limitation is addressed in the discussion section (Page 16, Lines 542–544, red words).

Revision: In addition, this study did not conduct follow-up tracking of participants' usage and improvements after the intervent

Round 2

Reviewer 3 Report

Comments and Suggestions for Authors

Authors have adequately answered all of my concerns.